# Are the Latent Representations of Foundation Models for Pathology Invariant to Rotation?

**Matouš Elphick**[1,2,3] (iD)                    MATOUS.ELPHICK@CRICK.AC.UK

**Guang Yang**[*3] (iD)                              G.YANG@IMPERIAL.AC.UK

**Samra Turajlic**[*1,2] (iD)                    SAMRA.TURAJLIC@CRICK.AC.UK

[1] *The Francis Crick Institute,* [2] *The Institute of Cancer Research,* [3] *Imperial College London*

**Editors:** Accepted for publication at MIDL 2025

## Abstract

Self-supervised foundation models for digital pathology encode small patches from H&E whole slide images into latent representations used for downstream tasks. However, the invariance of these representations to patch rotation remains unexplored. This study investigates the rotational invariance of latent representations across twelve foundation models by quantifying the alignment between non-rotated and rotated patches using mutual $k$-nearest neighbours and cosine distance. Models that incorporated rotation augmentation during self-supervised training exhibited significantly greater invariance to rotations. We hypothesise that the absence of rotational inductive bias in the transformer architecture necessitates rotation augmentation during training to achieve learned invariance. Code [1].

**Keywords:** computational pathology, oncology, foundation models, latent representations

## 1. Introduction

Digital pathology has enabled the use of deep learning on H&E whole slide images (WSIs). Foundation models (FMs), trained through self-supervised learning on large WSI datasets, generate compact latent representations which can be used downstream for tasks such as tumour detection and grading. However, as these models are adopted in practice, concerns have arisen about their robustness. While most studies on FMs for pathology evaluate their accuracy and ability to generalise across datasets, few have investigated their resilience to geometric transformations such as rotation. Ideally, such transformations should not affect the diagnostic features encoded in the latent representations, and understanding the impact of rotation is crucial, as invariance could enhance model reliability and robustness. However, the lack of rotational inductive bias in the transformer architecture suggests that achieving invariance may require rotation augmentation during training. This paper investigates the extent to which the latent representations of FMs remain invariant to rotation by quantifying the alignment between representations of images with and without rotation.

## 2. Methods

### 2.1. Dataset

For this study, the FMs were benchmarked using the TCGA-KIRC dataset (Akin et al., 2016), which contains 363 H&E WSIs. WSIs are read at $20\times$ magnification and converted

---

* Joint senior authorship

1. https://github.com/MatousE/rot-invariance-analysis

to RGB and HSV color spaces. Non-background regions in the HSV images are isolated through segmentation, noise reduction, and contour detection. The five largest contours are selected, and $256 \times 256$ pixel patches are extracted from their bounding boxes, ensuring at least 75% of pixels are foreground.

## 2.2. Models

In this study, we evaluated twelve FMs that have used self-supervised training on WSI datasets including: Conch (Lu et al., 2024), Hibou (base and large) (Nechaev et al., 2024), Kaiko (base and large) (ai et al., 2024), PathDino (Alfasly et al., 2024), Phikon (Filiot et al., 2023), Phikon 2 (Filiot et al., 2024), Prov-GigaPath (Xu et al., 2024), UNI (Chen et al., 2024), Virchow (Vorontsov et al., 2024) and Virchow 2 (Zimmermann et al., 2024).

## 2.3. Rotations

We applied rotations from $0°$ to $360°$ at $15°$ intervals to WSI patches and applied the FM to the patch at each rotation and extracted the final latent representation. The 'control' condition, with no rotation, serves as the baseline for comparison.

## 2.4. Metric

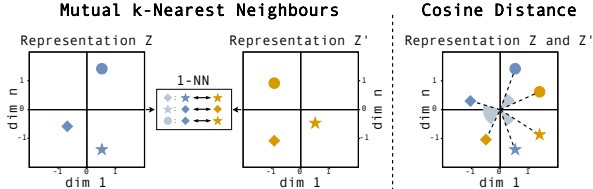

**Figure 1:** (Left) Representations are compared based on similarity of their $k$ nearest neighbors, here k = 1. (Right) Distance is measured by the cosine of the angle between representations.

To quantify the alignment between control and rotated latent representations, we used two metrics: mutual $k$-nearest neighbours (m-$k$NN) (Fig. 1 Left), which evaluates the proximity of similar images in latent space, and cosine distance, which measures how consistently images are encoded in the same direction (Fig. 1 Right) (Klabunde et al., 2024; Huh et al., 2024). These metrics are justified as H&E WSI patches of a similar type should not only be close in latent space but also map in similar directions, regardless of rotation. Let $X = \{x_i \in \mathbb{R}^{H \times W \times 3}\}_{i=1}^N$ represent a set of image patches extracted from WSIs, where $H$ and $W$ are the patch dimensions, and 3 is the number of channels. A model $f$ maps each patch to a latent representation, yielding $Z = \{z_i = f(x_i) \in \mathbb{R}^d\}_{i=1}^N$ where $d$ is the representation dimension. Given a rotation $\mathcal{R}$ which is applied to the input patches, the resulting representations are $Z' = \{z_i' = f(\mathcal{R}(x_i)) \in \mathbb{R}^d\}_{i=1}^N$. Now let $\mathcal{N}_k(z_i)$ be the set of $k$-nearest neighbours of $z_i \in Z$ and $\mathcal{N}_k(z_i')$ be the $k$-nearest neighbours of $z_i' \in Z'$, measured in the respective latent spaces with Euclidean distance. The m-$k$NN between $Z$ and $Z'$ is then defined:

$$m_{\text{NN}}^k(Z, Z') = \frac{1}{N} \sum_{i=1}^N \frac{|\mathcal{N}_k(z_i) \cap \mathcal{N}_k(z_i')|}{k}. \tag{1}$$

The cosine distance between $Z$ and $Z'$ is:

$$\text{Cosine Distance}(Z, Z') = \frac{1}{N} \sum_{i=1}^N \left(1 - \frac{z_i \cdot z_i'}{\|z_i\| \|z_i'\|}\right). \tag{2}$$

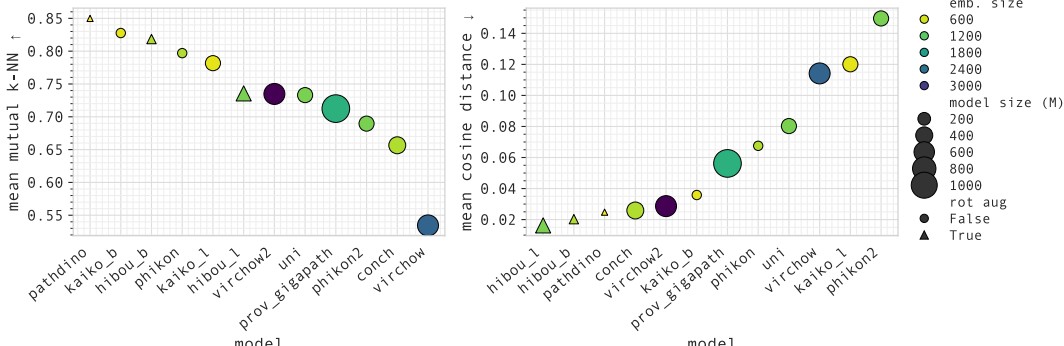

**Figure 2:** Graphs showing how model characteristics relate to representation invariance under rotation. (Left) Mean mutual $k$-nearest neighbour across rotations, with higher values indicating greater invariance. (Right) Mean cosine distance, where lower values indicate stronger alignment in latent space representations under rotation.

## 3. Results

In this study, FM invariance to H&E WSI patch rotation is evaluated using m-$k$NN and cosine distance. Latent representations were extracted for each patch at $15°$ increments across all models. m-$k$NN, with $k = 10$, and cosine distance were computed between non-rotated and rotated representations to quantify invariance. Fig. 2 shows the mean m-$k$NN and cosine distances across rotations. PathDino was the most invariant for m-$k$NN (0.85), while Virchow was the least (0.53). For cosine distance, Hibou-L showed the most invariance (0.016), whereas Phikon2 showed the least (0.145). Models were divided into two groups based on whether they used rotation augmentation during training. A t-test comparing these groups showed significant differences in both metrics: models trained with rotation augmentation achieved better scores for both cosine distance ($t = -8.88$, $p < 0.0001$) and m-$k$NN ($t = 6.91$, $p < 0.0001$). Fig. 3 presents a heatmap of m-$k$NN and cosine distances across models and rotations. Alignment was poorest at angles between cardinal rotations ($45°$, $135°$, $225°$, $315°$), likely due to slight differences in the patch corners introduced by rotation.

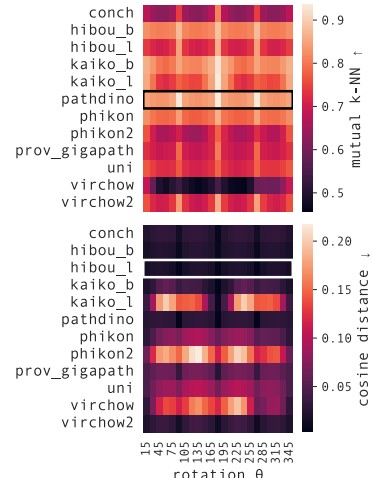

**Figure 3:** (Top) Mutual $k$-nearest neighbour across rotation angles, where higher values indicate greater invariance. (Bottom) Cosine distance across rotations, with lower values indicating greater latent representation alignment.

## 4. Conclusion

This study revealed that FMs for pathology exhibit varying degrees of rotational invariance, with models that use rotation augmentation during training being more invariant to rotation than those that do not. This result suggests that due to the transformers' lack of rotational inductive bias, rotation augmentation is necessary to achieve learned invariance. In future work, evaluation of rotation invariance could extend across a number of datasets and leverage several alignment metrics.

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
