# OpenReview forum: "Are the Latent Representations of Foundation Models for Pathology Invariant to Rotation?"
_MIDL.io/2025/Short_Papers — MIDL 2025 - Short Papers_

### Official Review · Reviewer_Jd7B · 2025-04-25

**Rating:** 5
**Confidence:** 4

**Summary:**

This paper presents a study to investigate whether12  foundation models (FM) for digital pathology are invariant to rotation. They extract image patches from H&E images, rotate them at varying degrees, and compute 2 metrics to quantify whether the FM representations are invariant to rotation. They found that models that had incorporated rotation augmentation during training were more invariant to rotation.

**Strengths:**

1. The motivation of the study is important, as objects in pathology images have no inherent orientation, and thus one should expect that a model analyzing such data should be invariant to rotations, i.e., we don't want the model predictions to change if the objects are oriented at differing angles.

2. The study looks at a large number (12) of FMs for pathology analysis and experiments are performed on a public dataset, enhancing reproducibility.

3. The study conclusions may be helpful for other researchers who are applying these models.

**Weaknesses:**

1. The study looked at quantifying the robustness of the representations to changing rotations, which is useful, but also it would have been nice to see and quantify the effects of what happens when the rotated images are used for different downstream prediction problems, since the effect on a downstream task (e.g., cell/tissue classification) is really what we may be interested in.

2. While I do think it is an important study to show how rotation affects different pathology FM, the overall conclusion is expected - that if models were trained with rotation, they are more robust to rotations. But again, the readers will now nicely see summarized which models can do better under rotation.

---

### Decision · Program_Chairs · 2025-05-01

Accept